# Adipose Tissue Macrophage Polarization Is Altered during Recovery after Exercise: A Large-Scale Flow Cytometric Study

Kyung-Wan Baek [1,2] , Ji Hyun Kim [3] , Hak Sun Yu [4,5,*,†] and Ji-Seok Kim [1,2,*,†]

1   Research Institute of Pharmaceutical Sciences, Gyeongsang National University,
    Jinju 52828, Republic of Korea; baekbo1218@gmail.com
2   Department of Physical Education, Gyeongsang National University, Jinju 52828, Republic of Korea
3   Department of Food Science and Nutrition, Gyeongsang National University, Jinju 52725, Republic of Korea;
    jihyunkim@gnu.ac.kr
4   Department of Parasitology and Tropical Medicine, Pusan National University School of Medicine,
    Yangsan 50612, Republic of Korea
5   Research Institute for Convergence of Biomedical Science and Technology, Pusan National University
    Yangsan Hospital, Yangsan 50612, Republic of Korea
*   Correspondence: hsyu@pusan.ac.kr (H.S.Y.); kjs7952@gnu.ac.kr (J.-S.K.)
†   These authors contributed equally to this work.

**Abstract:** We performed a large-scale flow cytometric analysis to determine whether M1 macrophage (M1Ø) and M2 macrophage (M2Ø) polarization in white adipose tissue (WAT) was altered immediately after exercise. Additionally, we comprehensively investigated the effects of obesity, exercise intensity, and recovery time on macrophage polarization in WAT. A single exercise bout of various intensities (ND, non-exercise control; -LIE, low-intensity exercise; -MIE, mid-intensity exercise; -HIE, high-intensity exercise) was performed by normal mice (ND) and obese mice (HFD). To confirm differences in M1Ø/M2Ø polarization in WAT based on the recovery time after a single exercise bout, WAT was acquired at 2 h, 24 h, and 48 h after exercise (total $n = 168$, 7 mice $\times$ 4 groups $\times$ 2 diets $\times$ 3 recovery time). The harvested WAT was immediately analyzed by flow cytometry, and macrophages were fluorescently labeled using F4/80, as well as M1Ø with CD11c and M2Øs with CD206. After a single bout of exercise, the M2Ø/M1Ø polarization ratio of WAT increases in both normal and obese mice, but differences vary depending on recovery time and intensity. Regardless of obesity, our findings showed that there could be a transient increase in M1Ø in WAT over a short recovery time (24 h) post-exercise (in ND-MIE, ND-HIE, and HFD-HIE). Furthermore, it was observed that the greater the exercise intensity in obese mice, the more effective the induction of M2Ø polarization immediately after exercise, as well as the maintenance of high M2Ø polarization, even after a prolonged recovery time.

**Keywords:** obesity; adipose tissue; macrophage; exercise; recovery

## 1. Introduction

Globally, the incidence of obesity and resulting public health challenges are growing at a rapid pace [1]. Obesity, which refers to excessive fat accumulation in the body, causes complications such as hyperlipidemia, hypertension, type 2 diabetes mellitus [2], and non-alcoholic fatty liver disease [3].

In obesity, chronic low-grade inflammation of the adipose tissue has been associated with metabolic diseases and organ tissue complications [4]. In patients with insulin resistance and obesity, white adipose tissue (WAT) is reportedly composed of a higher number of macrophages (M1 macrophages, M1Ø), along with an increased expression of pro-inflammatory mediators [5]; this phenomenon is closely related to metabolic dysfunction. Therefore, attenuation of WAT inflammation in obesity can be a possible treatment

strategy to avert metabolic and vascular complications associated with obesity [6,7]. Furthermore, it was recently revealed that macrophage polarization of WAT is closely related to inflammation and immunosenescence [8].

Previously, we have confirmed that relatively high-intensity training in mice improves macrophage polarization toward the anti-inflammatory phenotype in WAT [9]. In another study, calorie restriction with eccentric exercise reportedly upregulated anti-inflammatory macrophages (M2 macrophages, M2Ø) while downregulating M1Ø [10]. Based on these findings, exercise appears to be an important factor that can improve insulin resistance by decreasing M1Ø polarization while increasing M2Ø polarization in obese WAT. Furthermore, M2Ø polarization was reportedly elevated in the WAT of super-aged mice that performed lifelong spontaneous exercise; thus, it could be suggested that exercise can maintain energy homeostasis, thereby preventing inflamm-aging and reducing the risk of premature death [11].

Numerous studies have confirmed that exercise plays a vital role in the upregulation of M2Ø polarization [9–13], but several questions need to be clarified. Sustained exercise appears to induce M2Ø polarization; however, it remains unclear whether it can cause M2Ø polarization during acute response to exercise. Previous studies have confirmed that acute exercise increases the expression levels of anti-inflammatory genes in WAT [13]; however, an assessment of macrophage polarization by flow cytometry has not been performed. It is evident that the changes in macrophage polarization in adipose tissue, which are caused by acute exercise, are not continuous. Therefore, observing the polarization of macrophages in adipose tissue after exercise can help determine the sustainability of the macrophage polarization effect caused by exercise. Moreover, it can be used as evidence for further research on the levels of various adipokines, such as leptin, an appetite-regulating hormone, that change after exercise.

Herein, we conducted a large-scale flow cytometric analysis to confirm whether macrophage polarization in WAT was altered immediately after exercise. In addition, we aimed to comprehensively investigate the effects of obesity, exercise intensity, and recovery time on macrophage polarization in WAT.

## 2. Materials and Methods

### 2.1. Animals and Experimental Design

Six-week-old male C57BL/6 mice (*n* = 168) were purchased from Koatech (Gyeonggi-do, Republic of Korea); then, half (*n* = 84) received a normal diet (ND) (Rodent NIH-41 Open Formula Diet. Zeigler Bros., Inc., Gardners, PA, USA), and the remainder (*n* = 84) were fed a 60% high-fat diet (HFD) (D12492, Research Diets, New Brunswick, NJ, USA) for 8 weeks. The mice were housed in a room maintained at 24 °C with 50–60% relative humidity and a 12 h light/12 h dark cycle. ND-fed mice were divided into a control group (ND) that did not exercise and groups that performed low-intensity (ND-LIE), mid-intensity (ND-MIE), and high-intensity (ND-HIE) exercise. Furthermore, HFD-induced obese mice were divided into a control group (HFD), as well as low-intensity (HFD-LIE), mid-intensity (HFD-MIE), and high-intensity (HFD-HIE) exercise groups. All control and exercise groups were further assessed at 3 recovery time points (2 h, 24 h, and 48 h, each group) to confirm macrophage polarization according to recovery time (total *n* = 168, 7 mice × 4 groups × 2 diets × 3 recovery time; ND, *n* = 21; ND-LIE, *n* = 21; ND-MIE, *n* = 21; ND-HIE, *n* = 21; HFD, *n* = 21; HFD-LIE, *n* = 21; HFD-MIE, *n* = 21; and HFD-HIE, *n* = 21). Following inhalation anesthesia with $CO_2$ gas, the abdominal cavity was opened, and the epididymal WAT was harvested. All experimental protocols were approved by the Institutional Animal Care and Use Committee of Pusan National University (approval number: PNU-2018-1957).

### 2.2. Exercise Protocol

During the 8-week obesity induction period, all mice performed practice running once a week at the same exercise intensity on an animal treadmill (DJ-344; Daejong Instrument

Industry, Daejeon, Republic of Korea). After the 8-week obesity induction period, a single exercise bout of different intensities was performed with the same total amount of exercise (LIE, 12 m/min for 75 min; MIE, 15 m/min for 60 min; HIE, 18 m/min for 50 min), using the same animal treadmill that was employed during practice. Exercise intensity and volume settings were based on our previous study [9].

As a control, non-exercising animals were placed on an idle treadmill for the same time period. Before each exercise training session, all running mice were allowed a 5 min warm-up phase with a gradual increase in speed. To confirm the difference in macrophage polarization according to the recovery time, mice corresponding to the 48 h recovery time performed exercise 2 days prior to sacrifice; mice corresponding to the 24 h recovery time performed the exercise bout 1 day before sacrifice. On the day of sacrifice, mice corresponding to a recovery time of 2 h were subjected to exercise. The time of sacrifice time was matched to the recovery time after exercise, and the exercise group of the same intensity was sacrificed simultaneously.

*2.3. Flow Cytometry*

Flow cytometric analysis was performed based on previously described experimental methods [1,2]. Immediately after mice sacrifice, macrophage polarization was confirmed in live cells in WAT. Epididymal WAT from mice was rinsed in saline, minced into fine pieces, and digested for 1 h at 37 °C in HEPES buffer (pH 7.4) containing 0.5 g/L of type I collagenase (Sigma-Aldrich, St. Louis, MO, USA) and 2% dialyzed bovine serum albumin (BSA, Fraction V; Sigma-Aldrich). The digested WAT was passed through a mesh filter (100 μm) and fractionated via brief centrifugation ($282 \times g$, 1200 rpm). Our isolation protocol is a separation method that includes the stromal vascular fraction present in adipose tissue. Isolated adipocytes were counted using a Countess II Automated Cell Counter (Life Technologies, Carlsbad, CA, USA). The cell viability was 80–90%. After counting, the cells were centrifuged at 1200 rpm for 5 min and resuspended in FACS buffer (2 mM of EDTA, 0.5% BSA) at a concentration of $2 \times 10^6$ cells/mL. Cells were incubated in the dark at 4 °C in blocking antibody (20 μg/mL) on a bidirectional shaker for 30 min and then for an additional 50 min with fluorophore-conjugated primary antibodies or isotype control antibodies. Then, cells were stained with anti-F4/80-FITC, anti-CD11c-PE, and anti-CD206-APC (eBioscience, San Diego, CA, USA) according to the manufacturer's recommendations. After staining, the cells were subjected to flow cytometry (BD LSRFortessa™ X-20, BD Biosciences, San Jose, CA, USA; results were analyzed using FlowJo software version 10 (Treestar, Ashland, OR, USA).

*2.4. Statistical Analysis*

Statistical analysis was performed using GraphPad Prism 8.3.0 software (GraphPad Software, San Diego, CA, USA). Normality was verified using the Shapiro–Wilk test. Two-way ANOVA was performed for each recovery time (2 h, 24 h, and 48 h) to confirm the interaction between macrophage polarization and exercise intensity. Additionally, Sidak's multiple comparison test was performed to confirm the difference between M1Ø and M2Ø polarization within groups. Statistical significance was set at $p < 0.05$.

## 3. Results

*3.1. In Normal Mice, the Ratio of M2Ø/M1Ø Polarization in WAT Increases Temporarily after Relatevly HIE Acute Exercise*

In ND-fed mice, M2Ø polarization in WAT was significantly increased immediately after exercise (after 2 h) regardless of exercise intensity (a difference in significance level was observed) (Figure 1). No significant difference was observed in the M1Ø and M2Ø polarization after a recovery time exceeding 24 h. However, M1Ø polarization was significantly higher than M2Ø polarization in mice subjected to HIE (after 24 h; $p < 0.05$), with no significant difference detected after 48 h (Figure 1). In ND-LIE, the ratio of M2Ø polarization in WAT after a recovery time of 48 h was lower than immediately after exercise

and after a short recovery time (2 h vs. 48 h, 24 h vs. 48 h; $p < 0.05$). In ND-MIE and ND-HIE, the M2Ø polarization ratio in WAT was significantly lower in the recovery time after 24 h compared to the recovery time of 2 h post-exercise (2 h vs. 24 h). Also, there was no significant difference in the M2Ø polarization ratio in WAT between the recovery time of 24 h and the recovery time of 48 h post-exercise, but it was significantly lower than the recovery time after 2 h post-exercise.

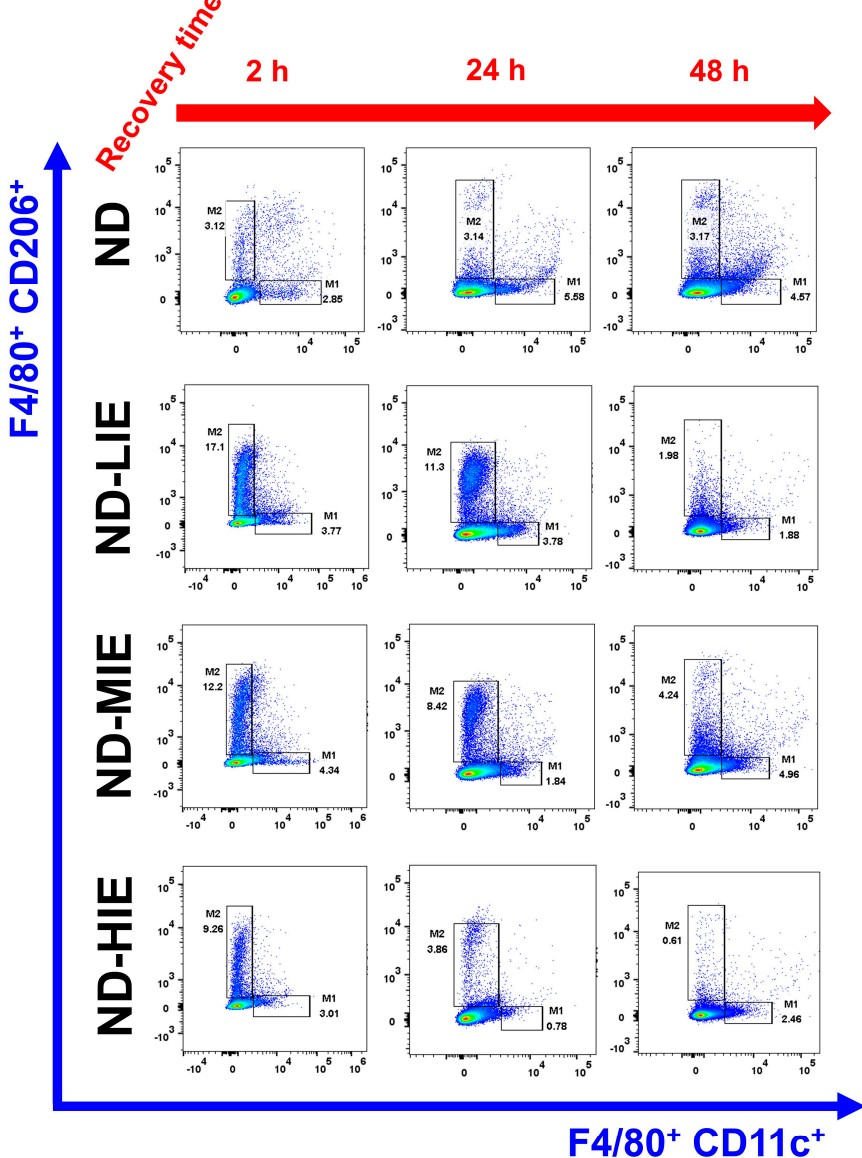

**Figure 1.** Differences in M1 macrophage (M1Ø) and M2 macrophage (M2Ø) polarization based on exercise intensity and recovery time in white adipose tissue (WAT) of normal mice. Representative flow cytometry plots of F4/80+ cells gated on macrophages. M1Ø-specific marker CD11c and M2Ø-specific marker CD206 subpopulations gated on F4/80+ cells. Please refer to Figure 3 for a statistical comparison.

*3.2. Macrophage Polarization in WAT of HFD-Induced Obese Mice Differed Based on Exercise Intensity Immediately after Exercise*

HFD-induced obese mice showed significantly higher M2Ø polarization than M1Ø polarization immediately after LIE (2 h after; $p < 0.05$) (Figure 2); however, 2 h after MIE, M1Ø polarization was significantly higher than M2Ø polarization ($p < 0.01$) (Figure 2). Immediately after HIE, M2Ø polarization was significantly higher than M1Ø polariza-

tion ($p < 0.0001$) (Figure 2), with a significance level higher than that of LIE (2 h HFD-LIE [$p < 0.05$] vs. 2 h HFD-HIE [$p < 0.001$]) (Figure 3F,H). In HFD-HIE, the M2Ø polarization ratio of WAT after 48 h of recovery time was significantly lower than that of 2 h post-exercise (2 h vs. 24 h; $p < 0.001$). However, the M2Ø polarization ratio of WAT was significantly higher in the recovery time of 48 h than in the recovery time of 24 h post-exercise (24 h vs. 48 h; $p < 0.001$) (Figure 4B).

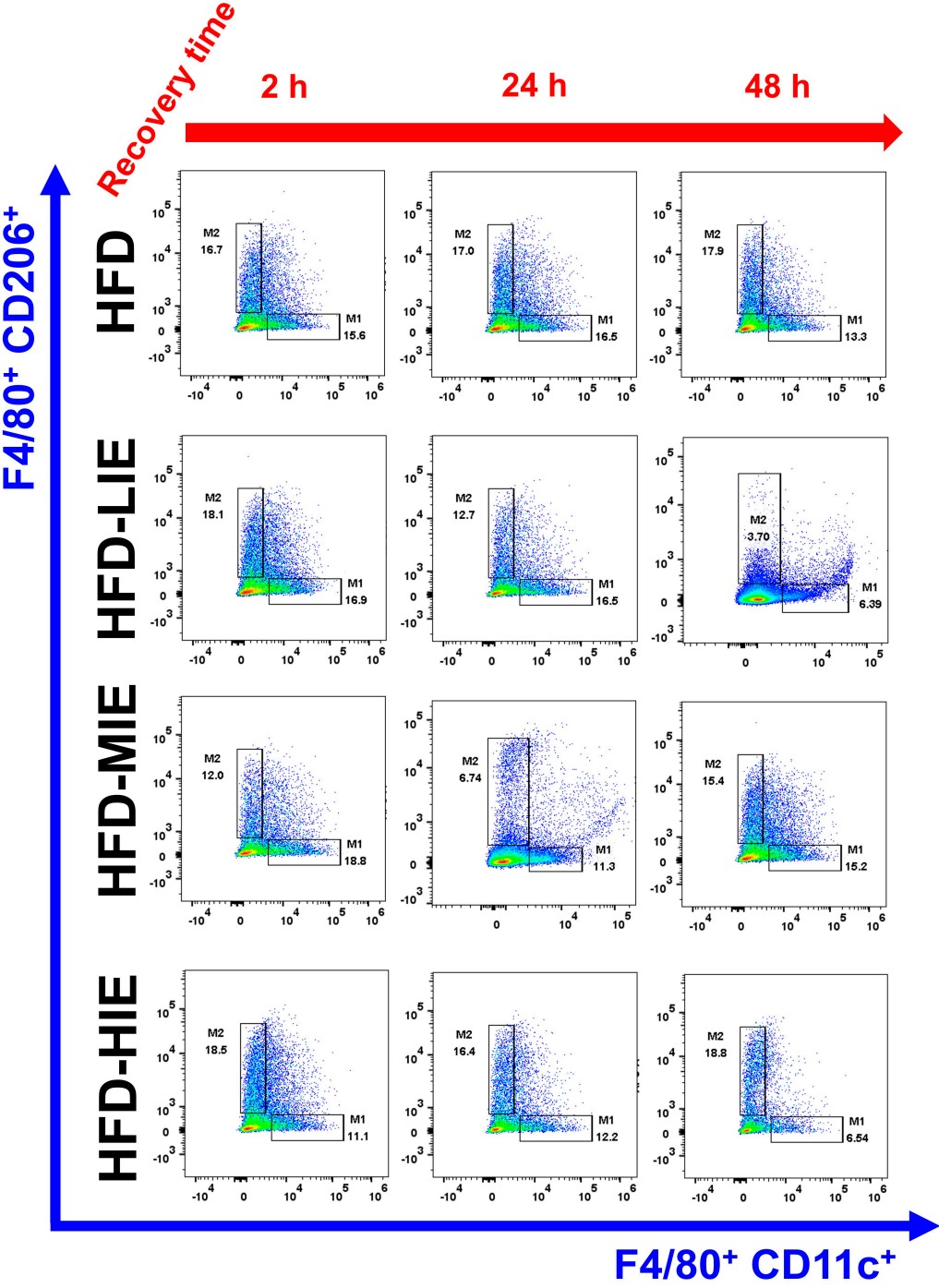

**Figure 2.** Differences in M1Ø and M2Ø polarization based on exercise intensity and recovery time in WAT of high-fat-diet-induced obese mice. Representative flow cytometry plots of F4/80+ cells gated on macrophages. M1Ø-specific marker CD11c and M2Ø-specific marker CD206 subpopulations gated on F4/80+ cells. Please refer to Figure 3 for a statistical comparison.

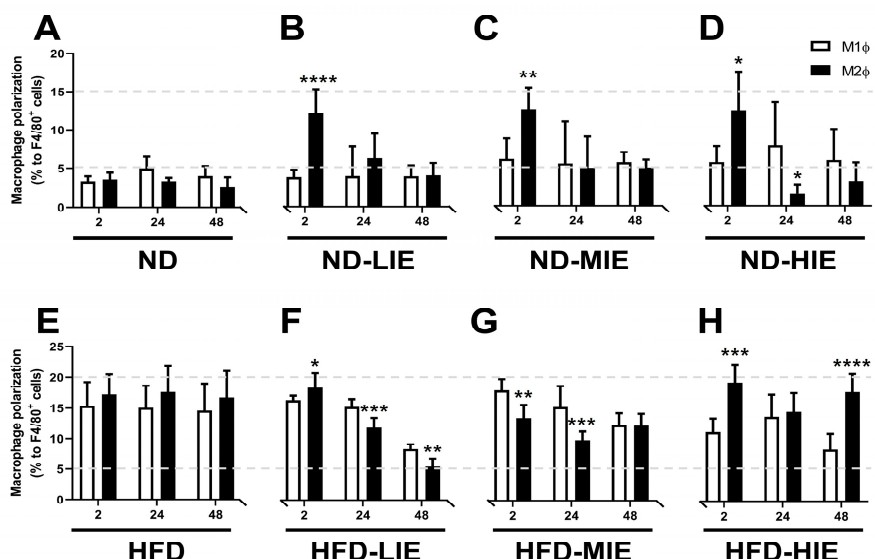

**Figure 3.** Statistical differences in M1Ø and M2Ø polarization ratio in WAT of all experimental groups. Changes in Ø polarization in WAT of mice on (**A**) normal diet (ND), (**B**) ND-low-intensity exercise (ND-LIE), (**C**) ND-moderate-intensity exercise, (**D**) ND-high-intensity exercise group, (**E**) high-fat diet (HFD), (**F**) HFD-low-intensity exercise, (**G**) HFD-moderate-intensity exercise, and (**H**) high-intensity exercise group according to recovery time after the exercise. Data values are presented as mean ± standard deviation (SD). * $p < 0.05$, ** $p < 0.01$ and *** $p < 0.001$, and **** $p < 0.0001$ M1Ø vs. M2Ø.

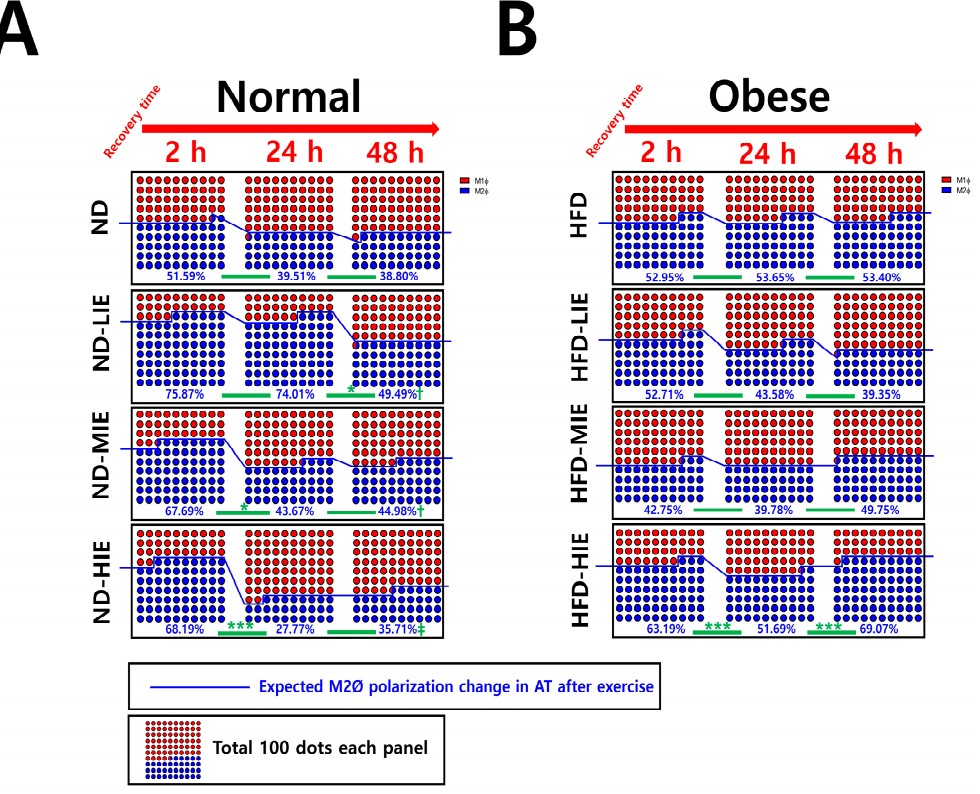

**Figure 4.** Estimation of changes in WAT M2Ø polarization ratio according to recovery time after exercise. (**A**) Changes in Ø polarization in WAT after exercise in normal mice, (**B**) Changes in Ø polarization in WAT after exercise in obese mice. Data values are presented as mean ± standard deviation (SD). * $p < 0.05$, and *** $p < 0.001$, comparison between groups connected by a line. † $p < 0.05$, and ‡ $p < 0.01$, vs. 2 h.

### 3.3. Relatively LIE Maintains a High M1Ø/M2Ø Polarization Ratio in WAT of HFD-Induced Obese Mice during Recovery

Despite a recovery time of 24 h post-exercise, both HFD-LIE and HFD-MIE groups showed significantly higher M1Ø polarization than M2Ø polarization in WAT (both $p < 0.001$) (Figures 2 and 3F,G). In particular, the HFD-LIE group demonstrated significantly higher M1Ø polarization than M2Ø polarization despite a 48 h recovery period ($p < 0.01$) (Figures 2 and 3F).

### 3.4. Relatively HIE Induces High M2Ø Polarization in WAT of HFD-Induced Obese Mice, Even after 48 h of Recovery

After a 48 h recovery period, the polarization of WAT macrophages did not appear to be significantly altered in ND-fed mice regardless of exercise intensity (Figures 1 and 3B–D). On assessing the WAT of obese mice after 48 h of recovery time, the HFD-LIE group revealed a high M1Ø/M2Ø polarization ratio ($p < 0.01$) (Figures 2 and 3F), but the HFD-HIE group demonstrated significantly higher M2Ø polarization than M1Ø polarization ($p < 0.0001$) (Figures 2 and 3H). Although HFD-HIE showed a higher M2Ø polarization ratio in WAT immediately after exercise (M1Ø vs. M2Ø, $p < 0.001$) (Figure 3H), there was no significant difference in M2Ø polarization in WAT after 2 h and 48 h post-exercise (Figure 4B). The purpose of including a non-exercise HFD group was to account for any inter-assay variations rather than changes in post-exercise recovery time. Our study showed no significant difference in the M2Ø/M1Ø polarization ratio of WAT in obese mice among three experiments (2 h, 24 h, and 48 h), indicating no inter-assay variations (Figure 3E).

## 4. Discussion

Previous reports have indicated that reducing the M1Ø/M2Ø polarization ratio of WAT by exercise training improves obesity and insulin resistance [1,3,4]. However, from a sports science perspective, as the amount and intensity of exercise increase, the risk of injury increases proportionally [5–8], and exercise without sufficient rest and nutritional supplementation can cause a decline in body function [9,10]. Therefore, exercise intensity and recovery time should be considered when exercising to maintain health.

In the present study, M2Ø polarization in WAT was effectively increased in ND mice immediately after exercise (after 2 h), regardless of exercise intensity (Figure 3B–D). However, it was observed that M1Ø polarization might increase temporarily during recovery time (after 24 h in ND-HIE) following HIE (Figure 3D). In ND mice, the M2Ø/M1Ø polarization ratio in WAT increased immediately after exercise and showed a tendency to reverse the ratio after a shorter recovery time as the relative exercise intensity increased (ND-LIE vs. ND-MIE or ND-HIE) (Figure 4A). Also, there was no significant change in ND-LIE after the recovery time of 24 h, but it showed a significantly lower trend after the recovery time of 48 h compared to the recovery time of 2 h and 24 h (Figure 4A). To summarize the results in ND mice, the increase in M2Ø polarization in WAT immediately after exercise was effective when the relative exercise intensity was low and the amount of exercise was the same (Figure 3A–D). However, since these results were obtained from normal mice, it is difficult to find a correlation with the improvement in obesity. Nevertheless, we examined the significance of the M1Ø/M2Ø macrophage polarization ratio changed by exercise in WAT of ND mice. The present study is the first report presenting these results, and hence, the precise underlying mechanism remains unclear. As WAT is a key organ regulating energy homeostasis, it can be speculated that M2Ø may be strongly associated with changes in the amount of energy consumed after exercise [11], increased excessive post-exercise oxygen consumption [11], and changes in appetite hormones [12]. Last but not least, it is important to point out that ND mice contain significantly less WAT than HFD mice. This may mean that the effect of WAT on energy homeostasis in ND mice may be significantly less than that in HFD mice. These various postulations need to be confirmed in further studies.

In HFD mice, the trend of M1Ø/M2Ø polarization in WAT revealed the effectiveness of relatively HIE. As shown in the flow cytometry analysis (Figure 3B), relatively LIE tended

to demonstrate significantly higher M2Ø polarization (vs. M1Ø, after 2 h in HFD-LIE) immediately after exercise. However, in general, M1Ø polarization tended to be higher than that of M2Ø immediately after exercise (HFD-MIE) and during recovery time (after 24 h and 48 h in HFD-LIE, after 24 h in HFD-MIE). As reported in our previous study, this study also confirmed that HIE could effectively induce high M2Ø polarization in the WAT of obese mice. In addition, relatively HIE increased M2Ø polarization after a long recovery period post-exercise (Figure 4B). Immediately after LIE in HFD mice, the M2Ø polarization ratio was significantly higher than M1Ø polarization; however, HFD-LIE mice revealed that M1Ø could increase with increasing recovery time. HFD-MIE mice presented a high M1Ø polarization ratio immediately after exercise and during recovery. There is currently no known evidence as to why this phenomenon occurs. However, on comparing HFD-LIE and HFD-MIE after a 48 h recovery period, HFD-MIE mice demonstrated relatively high M2Ø polarization; accordingly, HFD-MIE did not experience a lower exercise effect than HFD-LIE. In conclusion, assuming that the amount of exercise is the same, relatively HIE may be effective in increasing the M2Ø/M1Ø polarization ratio of WAT to improve obesity. The conclusion of this study, together with a study reporting that an increase in the M2Ø/M1Ø polarization ratio of obese WAT and an improvement in insulin resistance was shown with long-term (8 weeks) exercise training, suggests that relatively HIE can be effective in improving obesity [1].

On summarizing our findings based on the recovery time, a temporary increase in the M1Ø/M2Ø polarization ratio was observed, which appears to be primarily a response to exercise following a short recovery time (24 h) (in ND-MIE, ND-HIE, and HFD-HIE) (Figure 5).

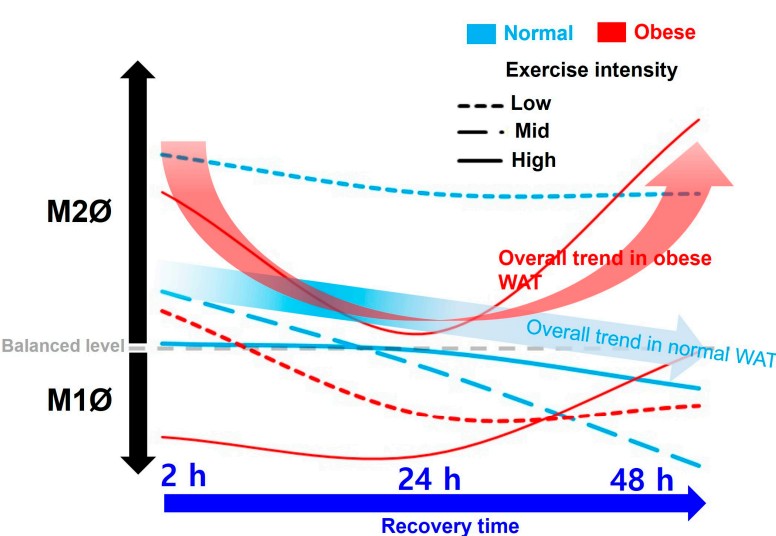

**Figure 5.** Summary of the effects of exercise intensity and obesity on changes in M1Ø/M2Ø polarization in WAT according to recovery time after exercise.

In obese mice, the higher the exercise intensity, the higher the M2Ø/M1Ø polarization ratio during the long recovery time (after 48 h). This finding differs from that observed in ND mice; as the intensity increases, M1Ø polarization increases significantly during the recovery time (especially ND-HIE). This result could be a reflex action to supplement the energy consumed post-exercise, as the body lacks excess energy in ND mice. As mentioned earlier, ND mice do not have any major problems with insulin resistance or energy homeostasis compared to HFD mice. Therefore, it would be more correct to view the macrophage polarization change in WAT in ND mice as a response to replenishing the energy consumed by exercise rather than as a recovery of energy homeostasis defect. In contrast, HFD mice exhibit excessive body fat accumulation and insulin resistance. When approached based on these facts, the fact that there was no significant difference in the ratio

of M1Ø and M2Ø in WAT 24 h after exercise in HFD-HIE and that the ratio of M2Ø/M1Ø polarization was high in WAT after 48 h in HFD-HIE were both facts can be seen as a series of processes occurring in the process of the recovery of energy homeostasis.

We faced several limitations during our study. Due to the high costs, time constraints, and manpower required for large-scale flow cytometry, various analyses could not be performed. Consequently, we were unable to identify the correlation between mouse phenotypic variables and blood biochemical variables, which is the biggest limitation of our study. Additionally, we did not confirm the size of the adipocytes and the amount of collected adipose tissue, which may have had an impact on the polarization of macrophages in adipose tissue. Although these factors would not have affected the results of flow cytometry, it would have been beneficial to identify and discuss them. Additionally, numerous studies have proven that exercise can induce angiogenesis in WAT [13–15]. Since exercise-induced "browning" of WAT is triggered by angiogenesis [16], it is also of great value to analyze the interaction between various angiogenesis-related genes and proteins and M1Ø/M2Ø polarization. We recommend that future studies take these limitations into account.

**5. Conclusions**

In this study, we report for the first time that the M2Ø/M1Ø polarization ratio of WAT increased immediately after exercise can change with recovery time, and exercise intensity can have a significant effect on this. We concluded that relative HIE could be effective for the improvement in obesity based on the M1Ø/M2Ø polarization ratio change pattern of WAT according to exercise intensity and recovery time.

**Author Contributions:** Conceptualization, K.-W.B. and H.S.Y.; methodology, K.-W.B.; software, K.-W.B.; validation, K.-W.B., J.H.K. and J.-S.K.; formal analysis, K.-W.B.; investigation, K.-W.B.; resources, H.S.Y. and J.-S.K.; data curation, K.-W.B.; writing—original draft preparation, K.-W.B.; writing—review and editing, K.-W.B. and J.H.K.; visualization, K.-W.B.; supervision, H.S.Y. and J.-S.K.; project administration, K.-W.B.; funding acquisition, K.-W.B. All authors have read and agreed to the published version of the manuscript.

**Funding:** This research was supported by the Basic Science Research Program through the National Research Foundation of Korea (NRF), funded by the Ministry of Education (NRF-2021R1I1A1A01044495 and NRF-2022S1A5A804884).

**Institutional Review Board Statement:** The animal study protocol was approved by the Institutional Animal Care and Use Committee of Pusan National University (approval number: PNU-2018-1957).

**Informed Consent Statement:** Not applicable.

**Data Availability Statement:** The original flow cytometric data of this study are restricted from being disclosed for our further studies. However, limited disclosure may be upon request with an appropriate explanation from the corresponding author.

**Conflicts of Interest:** The authors declare no conflicts of interest.

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
