# Peer review of "Adipose Tissue Macrophage Polarization Is Altered during Recovery after Exercise: A Large-Scale Flow Cytometric Study"

_cimb, doi:10.3390/cimb46020083_

Round 1

Reviewer 1 Report

Comments and Suggestions for Authors

1. Introduction should provide a better overview of the effect of acute exercise on macrophage polarization.

2. In introduction, it remains unclear what is this project proposing to add to the research. We already have data showing upregulation of M2-associated genes and M2 macrophage polarization with acute exercise. What do the authors propose to add to this? What benefit is it to know if the macrophage polarization is modified immediately or over a period of 48 hours?

3. Please provide reference to the amount of exercise (LIE vs MIE vs HIE) and the regimen used in this experiment.

4. Were the animals weighed after the 8-week obesity induction period to show significant differences in body weight? If so, please mention it. Did you also harvest more WAT from HFD-mice compared to ND-mice?

5. Result 3.1 title is confusing. If mentioned "Relatively HIE in normal mice increases the M1Ø/M2Ø polarization ratio in WAT", it suggests that there is polarization towards M1 macrophages with HIE categorically. Which does not seem to be true since the opposite is true within 2 hours of recovery and there is non-significant increase after 48 hours. 

6. Figure 5: Please correct M2Ø/M2Ø to M1Ø/M2Ø in the figure legend. Unclear which red line is mid vs. high intensity exercise in obese mice. Please label/re-design correctly.

7. Conclusions by the authors explaining differences in the polarization dynamics between ND and HFD mice seems unclear and vague. If it is suggested that the polarization seen in ND-HIE mice is seen as a reflexive correction of body energy levels, please explain how M1-polarization would address that. Additionally, if HFD-HIE polarization towards M2-marophages is a process of energy homeostasis, please further explain how this would be facilitated. The overall conclusion is not clear.

8. Please mention known limitations of this model including if the relatively HIE was enough cause significant changes in WAT with one acute episode.

Comments on the Quality of English Language

Few typos that need to corrected. No major issues. Please choose wording subheadings in results more accurately.

Author Response

  1. Introduction should provide a better overview of the effect of acute exercise on macrophage polarization.

Answer: Line 65 – 70, Thank you for providing your feedback on our manuscript. We appreciate your valuable comments. Previous studies have suggested that acute exercise-induced changes are temporary, but they have not been able to confirm it through flow cytometry. We have taken your advice and modified the manuscript to provide a better understanding of the significance of our research, particularly with regards to the effects of acute exercise. If you feel that the explanation is still not sufficient or if you have any further suggestions, please do not hesitate to let us know. We value your time and consideration.

  1. In introduction, it remains unclear what is this project proposing to add to the research. We already have data showing upregulation of M2-associated genes and M2 macrophage polarization with acute exercise. What do the authors propose to add to this? What benefit is it to know if the macrophage polarization is modified immediately or over a period of 48 hours?

Answer: Line 65 – 70, We have taken your advice and clarified the academic value that this study can provide.

  1. Please provide reference to the amount of exercise (LIE vs MIE vs HIE) and the regimen used in this experiment.

Answer: Line 98, Thank you for your advice. We noticed that citations for exercise intensity and volume were missing. Following your advice, we added this and cited our previous research

  1. Were the animals weighed after the 8-week obesity induction period to show significant differences in body weight? If so, please mention it. Did you also harvest more WAT from HFD-mice compared to ND-mice?

Answer: Thank you for your kind comments. We observed that after the period of inducing obesity in the animal, there was a clear increase in body weight and fat accumulation, which was visible to the naked eye. Due to the large number of mice and the extensive work required for flow cytometry, we were unable to perform various experiments. As a result, we submitted our research to this journal in the form of a Communication rather than an Article. We acknowledge that our study has both advantages and disadvantages, which we have clearly outlined. We hope you understand the various circumstances that impacted our research.

  1. Result 3.1 title is confusing. If mentioned "Relatively HIE in normal mice increases the M1Ø/M2Ø polarization ratio in WAT", it suggests that there is polarization towards M1 macrophages with HIE categorically. Which does not seem to be true since the opposite is true within 2 hours of recovery and there is non-significant increase after 48 hours. 

Answer: We have updated the title of the picture to align with your perspective. Thank you for your valuable advice.

  1. Figure 5: Please correct M2Ø/M2Ø to M1Ø/M2Ø in the figure legend. Unclear which red line is mid vs. high intensity exercise in obese mice. Please label/re-design correctly.

Answer: A typo has been corrected.

  1. Conclusions by the authors explaining differences in the polarization dynamics between ND and HFD mice seems unclear and vague. If it is suggested that the polarization seen in ND-HIE mice is seen as a reflexive correction of body energy levels, please explain how M1-polarization would address that. Additionally, if HFD-HIE polarization towards M2-marophages is a process of energy homeostasis, please further explain how this would be facilitated. The overall conclusion is not clear.

Answer: We believe your question is quite clear. As previously stated, the manuscript was submitted in communication format, rather than article format. To address your question, several additional studies will be necessary, which we are currently working on. It's important to understand that this study serves as a starting point. While our overall conclusion may not be entirely clear, we were able to confirm that exercise and exercise intensity do affect M1/M2 polarization in adipose tissue. Furthermore, we observed differences depending on the recovery time. I hope you can recognize the academic value of these findings. 

  1. Please mention known limitations of this model including if the relatively HIE was enough cause significant changes in WAT with one acute episode.

Answer: Line 283, Our manuscript mainly reports only the phenomenon. We reported that relative HIE causes significant changes in WAT, but the exact mechanism by which this occurs is difficult to understand from an immunological perspective to date. Our study has various limitations, which we are clearly aware of. We mentioned at the end of the discussion why we submitted it as a communication rather than an article and what the experimental limitations were. We will take your good opinions into consideration and improve them in our next research. We would appreciate it if you could understand these facts. 

Reviewer 2 Report

Comments and Suggestions for Authors

Baek et al report changes in macrophage polarization in white adipose tissue after different intensities of exercise. The authors employ mouse model with or without high fat feeding.

The study is extremely descriptive and contains no mechanistic incites. The abstract does not describe state of knowledge or clear research question. The methodology is not fully appropriate. How do authors define M1 and M2 macrophages with flow cytometry? What about other inflammatory cells.

The results are not well presented. There is not information regarding gating strategy. Some of the differences in figure 3, e.g. panel F are very small and most likely not significant. 

In summary, the work corresponds more to limited experiment and is descriptive having little merit. 

Comments on the Quality of English Language

Some sections of the manusrcipt is hard to understand because of poor language. 

Author Response

Baek et al report changes in macrophage polarization in white adipose tissue after different intensities of exercise. The authors employ mouse model with or without high fat feeding.

The study is extremely descriptive and contains no mechanistic incites. The abstract does not describe state of knowledge or clear research question. The methodology is not fully appropriate. How do authors define M1 and M2 macrophages with flow cytometry? What about other inflammatory cells.

The results are not well presented. There is not information regarding gating strategy. Some of the differences in figure 3, e.g. panel F are very small and most likely not significant. 

In summary, the work corresponds more to limited experiment and is descriptive having little merit. 

Answer: Line 283, We acknowledge your critical comments. Due to the limitations you mentioned, we decided not to submit our findings in the form of an article, but instead submitted it as a communication. We are fully aware of these limitations and have described them at the end of the discussion. We hope you can understand these points. Despite some controversial results, our study clearly shows that exercise, exercise intensity, and post-exercise recovery time have an impact on the polarization of macrophages in adipose tissue. We believe that reporting these results will be useful for future studies.

Reviewer 3 Report

Comments and Suggestions for Authors

The aim of this study was to examine the effect of obesity induced by high fat diet as well as the single boot of physical exercise on macrophage polarization toward the M1 or M2 phenotype using flow cytometry method. The animals were fed either regular or high fat diet and then were subjected to low-intensity, moderate-intensity or high-intensity exercise. White adipose tissue (WAT) samples were obtained 2, 24 or 48 hours after the exercise. The exercise was performed on the animal treadmill and exercise intensity (time and speed) was regulated. Epidydimal adipose tissue was collected under anesthesia and its cells were labelled with anti-F4/80, anti-CD11c, and anti-CD206 antibodies to detect total, M1 and M2 macrophages, respectively. They demonstrate that in normal diet fed mice M2 polarization increases immediately after the exercise regardless of its intensity. In high fat diet fed mice low-intensity exercise reduced the amount of both types of macrophages whereas high-intensity exercise had the opposite effect,

The topic and they results are of interest, however, there are also important concerns to be addressed.

1)     The composition of both diets should be reported. At least, percent of calories derived from carbohydrates, fat and proteins should be specified.

2)     All experiments were performed only in male mice. The results could be different in female mice. Using only male mice is a serious limitation of this study.

3)     Statistical analysis: was normality of data distribution verified to justify using parametric test (ANOVA)?

4)     Fig. 1D: it seems that without exercise (Fig. 1A) there is no difference between the amount of M1 and M2 macrophages, whereas 24 and 48 h after high-intensity exercise (Fig. 1D) M2 are less abundant than M1. What is the mechanism of this effect?

5)     While describing the results, the amount of macrophages belonging to the specific population at different time points after the exercise should always be compared to their number without exercise for better clarity. Comparing only M1 to M2 is not sufficient to analyze the results.

6)     Basic metabolic data of the animals should be presented including body weight, relative adipose tissue content, plasma lipids, glucose, insulin, adipokines. Without these data, the results are simply descriptive and do not allow to conclude much about their implications.

7)     Only one adipose tissue depot was used. It is unclear if the results may be extrapolated to other depots of visceral and/or to subcutaneous WAT.

Comments on the Quality of English Language

English language is fine; no language revision is required.

Author Response

Reviewer #3

The aim of this study was to examine the effect of obesity induced by high fat diet as well as the single boot of physical exercise on macrophage polarization toward the M1 or M2 phenotype using flow cytometry method. The animals were fed either regular or high fat diet and then were subjected to low-intensity, moderate-intensity or high-intensity exercise. White adipose tissue (WAT) samples were obtained 2, 24 or 48 hours after the exercise. The exercise was performed on the animal treadmill and exercise intensity (time and speed) was regulated. Epidydimal adipose tissue was collected under anesthesia and its cells were labelled with anti-F4/80, anti-CD11c, and anti-CD206 antibodies to detect total, M1 and M2 macrophages, respectively. They demonstrate that in normal diet fed mice M2 polarization increases immediately after the exercise regardless of its intensity. In high fat diet fed mice low-intensity exercise reduced the amount of both types of macrophages whereas high-intensity exercise had the opposite effect,

The topic and they results are of interest, however, there are also important concerns to be addressed.

Answer: Thank you for taking the time to provide us with your review. We appreciate your feedback and recognize the limitations of our study. Due to these limitations, we have decided to publish our findings as a communication rather than a full article. We have also clearly described the limitations of our study in the discussion section, specifically on line 283. While our research may have some shortcomings, we believe it still provides valuable insights and direction. We hope that you can see the value in reporting our results as they are. Thanks again for your review.

  • The composition of both diets should be reported. At least, percent of calories derived from carbohydrates, fat and proteins should be specified.

Answer: We recorded the catalog numbers and company names of the regular and high-fat diets that we used in our study. The most widely used commercial general feed and a 60% super fat diet were used to induce obesity, insulin resistance, and fatty liver due to their high energy efficiency compared to regular feed. The high-fat feed is commonly used for this purpose, as its effectiveness has been established in previous studies. Instead of displaying the calorie content of these feeds in a separate table, we found it more efficient to provide information on their ingredients. Therefore, we included this information and also added details about the missing normal diet.

  • All experiments were performed only in male mice. The results could be different in female mice. Using only male mice is a serious limitation of this study.

Answer: We understand and acknowledge your point that changes in macrophage polarization in adipose tissue can have different effects in male and female mice (https://journals.aai.org/jimmunol/article/206/1/141/107934).

However, it is important to note that males tend to exhibit a greater response than females when it comes to researching macrophage polarization changes due to exercise. As a result, males are primarily used for research related to macrophage polarization. Also, please note that the purpose of our study is not to compare gender differences.

  • Statistical analysis: was normality of data distribution verified to justify using parametric test (ANOVA)?

Answer: Line 130, The normal distribution of data was confirmed by conducting the Shapiro-Wilk test, which is explained in the statistical methods.

  • 1D: it seems that without exercise (Fig. 1A) there is no difference between the amount of M1 and M2 macrophages, whereas 24 and 48 h after high-intensity exercise (Fig. 1D) M2 are less abundant than M1. What is the mechanism of this effect?

Answer: Line 284, We apologize for not being able to provide a satisfactory answer to your question. The reason for this is that there were certain limitations with the study which prevented us from conducting some experimental methods. Unfortunately, due to these limitations, we are unable to explain the mechanism in detail. As we mentioned earlier, we submitted this study as a communication rather than an article, acknowledging the limitations of the study. We appreciate your valuable comments and suggestions and they will help us in our future research endeavors. Thank you.

5)     While describing the results, the amount of macrophages belonging to the specific population at different time points after the exercise should always be compared to their number without exercise for better clarity. Comparing only M1 to M2 is not sufficient to analyze the results.

Answer: Line 284, We apologize for not being able to provide a satisfactory answer to your question. The reason for this is that there were certain limitations with the study which prevented us from conducting some experimental methods. Unfortunately, due to these limitations, we are unable to explain the mechanism in detail. As we mentioned earlier, we submitted this study as a communication rather than an article, acknowledging the limitations of the study. We appreciate your valuable comments and suggestions and they will help us in our future research endeavors. Thank you.

  • Basic metabolic data of the animals should be presented including body weight, relative adipose tissue content, plasma lipids, glucose, insulin, adipokines. Without these data, the results are simply descriptive and do not allow to conclude much about their implications.

Answer: Line 284, We apologize for not being able to provide a satisfactory answer to your question. The reason for this is that there were certain limitations with the study which prevented us from conducting some experimental methods. Unfortunately, due to these limitations, we are unable to explain the mechanism in detail. As we mentioned earlier, we submitted this study as a communication rather than an article, acknowledging the limitations of the study. We appreciate your valuable comments and suggestions and they will help us in our future research endeavors. Thank you.

  • Only one adipose tissue depot was used. It is unclear if the results may be extrapolated to other depots of visceral and/or to subcutaneous WAT.

Answer: Line 90, We used epididymal fat in our study as it is an adipocyte that is known to be useful in studying macrophage polarization. While there may be similarities between visceral fat and subcutaneous fat, it is not accurate to say that they have opposite tendencies. Our study is an exception, and due to various limitations, it is difficult to classify different types of local organizations. Those who have conducted similar experiments will understand this situation. If we were to analyze a large number of mice by dividing them into visceral fat and subcutaneous fat, the time and cost required for our processor flow cytometry would double. Moreover, there is a possibility that the experiment may fail if not performed properly. Please take into account the size and amount of work we have put into our research.

Reviewer 4 Report

Comments and Suggestions for Authors

The study investigated the effect of obesity and exercise on the polarization of macrophages in white adipose tissue. The authors found that greater exercise intensity in obese mice induces greater M2 polarization after exercise, which persists even during the recovery period.

The study is well designed, the presented results are interesting, and well documented. The study is focused on an interesting topic and brings new findings. On the contrary, English is not good and must be improved. Many sentences are difficult to understand.

Before accepting of study results and publishing them, several facts should be clarified:

Abstract:

-Information about the method can be abbreviated (e.g. division of animals into subgroups). On the contrary, the results should be described in more detail. E.g. sentence line 26-27: "Following a single bout of exercise, the M1Ø/M2Ø polarization ratio in WAT differed between normal and obese mice", does not provide sufficient specific information.

Materials and Methods section:

-The exact number of animals in each group should have been stated.

-The authors should explain on the basis of which the intensity and duration of the exercise were simultaneously changed in the exercise cycle. It is not so obvious whether the results are influenced by the intensity or the duration of the exercise. Please explain.

-Analysis of macrophage polarization using flow cytometry was performed on isolated adipocytes. But macrophages are present in the stromavascular fraction, which is not included in the analysis in this study. Please explain.

Results::

-Figure 5 is somewhat confusing and so is its description in the discussion text (lines 254-256).There is probably an error in the labelling of the lines, since the results for the light and highly obese groups cannot be distinguished, and the recovery period should be supplemented with time.

Discussion:

-Many variable parameters are monitored in the study, and due to this, the discussion is difficult to summarize, which is also contributed to by poor English. Instead of repeating the results , possible mechanisms influencing the recruitment of macrophages into adipose tissue and their polarization should be mentioned in the discussion.

-It should be noted that the results in recovery time can also be influenced by the size of adipocytes. The relationship between adipocyte size and the percentage of macrophages in adipose tissue has been demonstrated in both animal and human studies. Exercise can differentially affect the number or polarization of macrophages: in lean mice, exercise can reduce adipocyte size, and hypertrophy and possibly hyperplasia can occur during recovery. In contrast, physical exercise may not have such effects in obese mice.

Author Response

The study investigated the effect of obesity and exercise on the polarization of macrophages in white adipose tissue. The authors found that greater exercise intensity in obese mice induces greater M2 polarization after exercise, which persists even during the recovery period.

The study is well designed, the presented results are interesting, and well documented. The study is focused on an interesting topic and brings new findings. On the contrary, English is not good and must be improved. Many sentences are difficult to understand.

Before accepting of study results and publishing them, several facts should be clarified:

Abstract:

-Information about the method can be abbreviated (e.g. division of animals into subgroups). On the contrary, the results should be described in more detail. E.g. sentence line 26-27: "Following a single bout of exercise, the M1Ø/M2Ø polarization ratio in WAT differed between normal and obese mice", does not provide sufficient specific information.

Answer: Line 26, We have taken into account your feedback and considered various methods to improve the overview provided. However, due to the structure of our study, it would have been impossible to mention all statistical differences in terms of diet, exercise intensity, and recovery time without exceeding the length of the abstract. We hope you understand the nature of our research and we have tried our best to convey our thoughts as clearly as possible.

Materials and Methods section:

-The exact number of animals in each group should have been stated.

Answer: Line 89, Following your advice, I entered the number of animals in each group. Entering the number of animals per group was difficult due to the study's design. Please bear with us.

-The authors should explain on the basis of which the intensity and duration of the exercise were simultaneously changed in the exercise cycle. It is not so obvious whether the results are influenced by the intensity or the duration of the exercise. Please explain.

Answer: Line 284, We apologize for not being able to provide a satisfactory answer to your question. The reason for this is that there were certain limitations with the study which prevented us from conducting some experimental methods. Unfortunately, due to these limitations, we are unable to explain the mechanism in detail. As we mentioned earlier, we submitted this study as a communication rather than an article, acknowledging the limitations of the study. We appreciate your valuable comments and suggestions and they will help us in our future research endeavors. Thank you.

-Analysis of macrophage polarization using flow cytometry was performed on isolated adipocytes. But macrophages are present in the stromavascular fraction, which is not included in the analysis in this study. Please explain.

Answer: Line 119, We are impressed with your understanding. The manuscript states that our protocol polarizes macrophages and separates the stromal vascular fraction.

Results::

-Figure 5 is somewhat confusing and so is its description in the discussion text (lines 254-256).There is probably an error in the labelling of the lines, since the results for the light and highly obese groups cannot be distinguished, and the recovery period should be supplemented with time.

Answer: We appreciate your feedback. We aimed to present the change in macrophage polarization ratio based on exercise intensity and recovery period in a clear and concise way. To achieve this, we evaluated several methods and selected the most effective one, which is displayed in Figure 5 (Time display has been added to Figure 5). Despite our best efforts, we acknowledge that it may be difficult for readers to understand. We did consider excluding Figure 5 altogether, but if you feel it is unnecessary, we can remove it. Line 254 – 256, One sentence was modified and unnecessary sentences were deleted to simplify it.

Discussion:

-Many variable parameters are monitored in the study, and due to this, the discussion is difficult to summarize, which is also contributed to by poor English. Instead of repeating the results , possible mechanisms influencing the recruitment of macrophages into adipose tissue and their polarization should be mentioned in the discussion.

Answer: Line 254, Thank you for your kind comments. However, we regret to inform you that we were unable to perform a variety of experiments due to the large number of mice and extensive work required for flow cytometry. This also makes it difficult to explain various mechanisms. Therefore, we submitted our findings to this journal in the form of a communication rather than a paper. We acknowledge that our study has both strengths and weaknesses, and we have outlined these clearly. We hope you understand the various circumstances that influenced our research. Once this manuscript is approved, we will review it by experts, proofread it in English, and revise it.

-It should be noted that the results in recovery time can also be influenced by the size of adipocytes. The relationship between adipocyte size and the percentage of macrophages in adipose tissue has been demonstrated in both animal and human studies. Exercise can differentially affect the number or polarization of macrophages: in lean mice, exercise can reduce adipocyte size, and hypertrophy and possibly hyperplasia can occur during recovery. In contrast, physical exercise may not have such effects in obese mice.

Answer: Line 254, Thank you for your kind comments. However, we regret to inform you that we were unable to perform a variety of experiments due to the large number of mice and extensive work required for flow cytometry. This also makes it difficult to explain various mechanisms. Therefore, we submitted our findings to this journal in the form of a communication rather than a paper. We acknowledge that our study has both strengths and weaknesses, and we have outlined these clearly. We hope you understand the various circumstances that influenced our research.

Round 2

Reviewer 1 Report

Comments and Suggestions for Authors

Thank you for the edits and clarification

Author Response

We have made improvements to our manuscript based on your valuable review. We have addressed all the missing parts highlighted by the academic editor (Line 295).

Reviewer 3 Report

Comments and Suggestions for Authors

The manuscript has been revised according to the reviewers' comments. I have no further concerns.

Author Response

(The authors gave the same response as above.)
